# Prevalence of surgically correctable conditions among children in a mixed urban-rural community in Nigeria using the SOSAS survey tool: Implications for paediatric surgical capacity-building

Adesoji O. Ademuyiwa[1,2]*, Tinuola O. Odugbemi[3], Christopher O. Bode[1,2], Olumide A. Elebute[1,2], Felix M. Alakaloko[2], Eyitayo O. Alabi[1,4], Olufemi Bankole[1,5], Oluwaseun Ladipo-Ajayi[2], Justina O. Seyi-Olajide[2], Babasola Okusanya[6,7], Ogechi Abazie[8], Iyabo Y. Ademuyiwa[8], Amanda Onwuka[9], Tu Tran[10,11], Ayomide Makanjuola[2], Shailvi Gupta[12], Riinu Ots[13], Ewen M. Harrison[13], Dan Poenaru[14], Benedict C. Nwomeh[15]

1 Department of Surgery, Faculty of Clinical Sciences, College of Medicine, University of Lagos, Lagos, Nigeria, 2 Paediatric Surgery Unit, Department of Surgery, Lagos University Teaching Hospital, Idi Araba, Lagos, Nigeria, 3 Department of Community Health and Primary Care, Faculty of Clinical Sciences, College of Medicine, University of Lagos, Lagos, Nigeria, 4 Department of Orthopaedics and Trauma, Lagos University Teaching Hospital, Idi Araba, Lagos, Nigeria, 5 Neurosurgery Unit, Department of Surgery, Lagos University Teaching Hospital, Idi Araba, Lagos, Nigeria, 6 Department of Obstetrics and Gynaecology, Faculty of Clinical Sciences, College of Medicine, University of Lagos, Lagos, Nigeria, 7 Department of Obstetrics and Gynaecology, Lagos University Teaching Hospital, Idi Araba, Lagos, Nigeria, 8 Department of Nursing Science, Faculty of Clinical Sciences, College of Medicine, University of Lagos, Lagos, Nigeria, 9 Centre for Surgical Outcomes Research, and Centre for Innovation in Paediatric Practice, The Research Institute at Nationwide Children's Hospital, Columbus, Ohio, United States of America, 10 SOSAS Uganda, Duke University Division of Global Neurosurgery, Durham, North Carolina, United States of America, 11 University of Minnesota Medical School, Minneapolis, Minnesota, United States of America, 12 University of California San Francisco East Bay; Surgeons Overseas; San Francisco, California, United States of America, 13 Department of Surgery, University of Edinburgh, Edinburgh, United Kingdom, 14 McGill University Health Centre and Montreal Children's Hospital, Montreal, Canada, 15 Nationwide Children's Hospital, Columbus, Ohio, United States of America

* aoademuyiwa@cmul.edu.ng

## Abstract

### Background

In many low- and middle-income countries, data on the prevalence of surgical diseases have been derived primarily from hospital-based studies, which may lead to an underestimation of disease burden within the community. Community-based prevalence studies may provide better estimates of surgical need to enable proper resource allocation and prioritization of needs. This study aims to assess the prevalence of common surgical conditions among children in a diverse rural and urban population in Nigeria.

### Methods

Descriptive cross-sectional, community-based study to determine the prevalence of congenital and acquired surgical conditions among children in a diverse rural-urban area of

**Data Availability Statement:** Data are available from Dryad: https://doi.org/10.5061/dryad. m37pvmcxp.

**Funding:** AOA received CRC 2016/030, University of Lagos Central Research Committee Grant. www. unilag.edu.ng. The funders had no role in study design, data collection and analysis, decision to publish, or preparation of the manuscript.

**Competing interests:** The authors have declared that no competing interests exist.

Nigeria was conducted. Households, defined as one or more persons 'who eat from the same pot' or slept under the same roof the night before the interview, were randomized for inclusion in the study. Data was collected using an adapted and modified version of the interviewer-administered questionnaire—Surgeons OverSeas Assessment of Surgical Need (SOSAS) survey tool and analysed using the REDCap web-based analytic application.

## Main results

Eight-hundred-and-fifty-six households were surveyed, comprising 1,883 children. Eighty-one conditions were identified, the most common being umbilical hernias (20), inguinal hernias (13), and wound injuries to the extremities (9). The prevalence per 10,000 children was 85 for umbilical hernias (95% CI: 47, 123), and 61 for inguinal hernias (95% CI: 34, 88). The prevalence of hydroceles and undescended testes was comparable at 22 and 26 per 10,000 children, respectively. Children with surgical conditions had similar sociodemographic characteristics to healthy children in the study population.

## Conclusion

The most common congenital surgical conditions in our setting were umbilical hernias, while injuries were the most common acquired conditions. From our study, it is estimated that there will be about 2.9 million children with surgically correctable conditions in the nation. This suggests an acute need for training more paediatric surgeons.

## Introduction

One of the key messages of the Lancet Commission on Global Surgery (LCoGS) is that surgery is an essential part of healthcare delivery. [1] Therefore, surgical care should be an integral part of the healthcare plan and policy of any nation state. In most low- and middle-income countries (LMICs) data on the surgical prevalence of disease have been mainly derived from hospital-based studies [2–4]. Such data are potentially deficient in several aspects. [5] First, they may represent only a fraction of the actual disease prevalence in the community. Second, such data may be skewed in favour of services available in participating hospitals. Third, access to the hospital may be limited due to factors such as financial, geographic and time constraints; hence a certain proportion of the community is not included in these studies. Finally, such studies do not account for people in the community without a disease at the time of the study but who might have had it previously or are at risk of developing it later. Consequently, such studies have been of limited use in establishing policies that will adequately serve entire populations. In Nigeria, for instance, the authors are not aware of any population-based studies to determine the prevalence of surgical conditions among children.

The aim of this study was to determine the prevalence of surgical conditions in children through a household survey in the diverse rural-urban Ikorodu area of Lagos state, using a modified Surgeons OverSeas Assessment of Surgical Needs (SOSAS) interviewer-administered questionnaire, and appraise the adequacy of the surgical workforce for the estimated disease prevalence.

## Materials and methods

### Setting

This study was conducted in the Ikorodu community of Lagos state. Ikorodu is one of the 20 Local Government Areas (LGAs) of Lagos state and is classified as one of the four rural LGAs. Ikorodu LGA had a projected population size of 733,927 persons in 2016. This site was selected because of its blend of neo-urbanization with an aquatic/agrarian rural nature which typifies most of the Nigerian population.

### Community engagement

Appropriate community entry and mobilization was carried out, and with the help of community leaders and community focal persons there was adequate community participation. Adequate community mobilization and participation was achieved by the SOSAS team meeting with the Medical Officer of Health and getting permission to carry out the study in the Local Government. The team also had meetings and sought support from community leaders/representatives (traditional rulers, community development associations representatives, women and youth leaders) of the selected wards to inform them about the nature and purpose of the study. The leaders and community focal persons assisted with mobilizing the community and these encouraged participation by community members.

### Ethical considerations

Ethical approval was obtained from the Health Research and Ethics Committee of the Lagos University Teaching Hospital (LUTHHREC) (number ADM/DCST/HREC/APP/719). Community consent was given by the traditional leaders and the Community Development Association. Informed consent was also obtained from each participant above 18 years of age and from the parents or guardians of all children. Assent was sought from each child aged 13–17 years before the child was interviewed or physically examined.

### Study design and population

This was a descriptive cross-sectional community-based study among children in Ikorodu LGA of Lagos State. Each household recruited had no intention to relocate out of Ikorodu LGA in the forthcoming 12 months following the onset of the study. A household was defined as one or more persons 'who ate from the same pot', slept under the same roof the night before the interview and the members considered themselves a family unit with one member collectively recognized as the head/representative of the household [6].

### Sample size determination

A sample size of 3,715 participants was determined using the formula: n = $z^2$pq/$d^2$, where z was the standard normal scale set at 95% CI, d was the acceptable margin of error of our population proportion set at 1%, and p was the estimated prevalence of surgical conditions in the population (population proportion) set at 7.3% (based on a similar study done in Sierra Leone) [7]. Two corrections were further accounted for to achieve the final sample size: first, 10% was added for assumed non-response, and a design effect of 1.3 was used to adjust for clustering based on the proposed multistage sampling design.[6] Approximately 620 households were aimed for in the sample based on known average household sizes [8].

## Sampling technique

A multistage sampling technique was carried out to recruit respondents.

*Stage 1*: *Selection of wards*. Two wards (a rural and an urban designated ward) each from the six zones/council areas were selected by simple random sampling (balloting).

*Stage 2*: *Selection of streets/settlements*: a list of streets/settlements in the selected wards was obtained from the LG office. One street/settlement was selected per ward using a random number calculator.

*Stage 3*: *Selection of household*. All households in residential buildings within each selected settlement were recruited for the study. All children within each household were recruited for the study.

## Data collection tool

Data was collected using an adapted and modified version of an interviewer-administered questionnaire, the Surgeons OverSeas Assessment of Surgical Need (SOSAS) survey tool [6,7]. This survey tool was developed to assess the burden of surgical conditions in a population by the US-based non-governmental organization Surgeons OverSeas (SOS). SOSAS was designed using the Demographic and Health Surveys (DHS), the WHO Guidelines for Conducting Community Surveys for Injuries and Violence, and a survey tool for road traffic injuries. This tool was revised to measure the prevalence of surgically treatable conditions, and subsequently evaluated and adapted by the SOS International Surgical Research Group. This group consisted of over 46 experts both from LMICs and high-income countries with global surgical interests and expertise. The survey tool is open-source and the developer permits modification of the survey tool if pilot testing or research shows the necessity [9]. SOSAS has not been validated in Nigeria but was initially deployed in Sierra Leone [6], then replicated in Nepal, Rwanda and Uganda [10–12]. This study utilized a modified version of the SOSAS survey tool termed the SOSAS-NST (Nigeria Survey Tool). Permission to use and modify the SOSAS survey tool was granted by the developer.

The full questionnaire included a socio-demographic component, adult, women's health and child questionnaires, and a physical examination component. This study focuses on the results from the child questionnaire and physical exam component. The former evaluated the health state/ illness details in the prior 12 months of each child in the household, the history and management of congenital and acquired surgical problems affecting face, head, neck; chest/breast; back; abdomen; groin, genitalia, buttocks; and extremities (upper and lower limbs). Respondents who identified a condition for their children over the course of the interview completed a visual portfolio and physical exam. The visual portfolio is a compilation of pictures depicting surgical conditions that can be identified by parents or guardians as being similar to the condition present in their children or wards. The 13 conditions included in the visual portfolio were agreed through an e-Delphi process by paediatric surgeons across Africa [13]

## Data collection technique

House-to-house interviews were carried out by 5 teams led and supervised by SOSAS Nigeria members. Each team consisted of at least a clinical facilitator, a field assistant and 4 trained interviewers.

The interviewers were pre-internship medical graduates who underwent a screening process for recruitment following interest in an advert placed by the SOSAS Nigeria team. They all completed two 3-day training workshops approximately 2 weeks apart, on the purpose of the study, the use of the SOSAS survey tool using a training manual, the process of selecting

households and conducting interviews, and on ways to handle possible issues that may arise in the field.

Data was collected using the SOSAS survey questionnaire on an electronic/ mobile device and hand-written forms (as back-up). Each household interview was carried out by a designated team, in an agreed area with considerations for privacy of participants as well as comfort and safety of the team.

Prior to physical examination, a visual portfolio for the congenital conditions being evaluated was shown to parents or guardians and they were asked to point to conditions that were present in their children or wards. After the documentation of the findings using the visual portfolio, a different team undertook the physical examination of all household members. This team included at least one medical practitioner and adequate privacy was ensured throughout the examination. All children were examined with the consent of, and in the presence of one of the parents/guardians, for female household members at least one female research team member was always present. Households or household members who were not available on the first visit required the team revisiting. At the end of each interview the data was checked for completion and errors. Debriefing took place at the end of each day.

## Data management and analysis

Data was checked daily before uploading to a central server and validated by a statistician and lead investigator before being locked. The confidentiality of the data was guaranteed by the statistician and the lead investigator.

Data collection and management was done using the Research Electronic Data Capture system (REDCap, https://projectredcap.org/) [14]. Prevalence and variance estimation took into account the stratification of the population by zone and clustering across 13 wards. Weights were applied to adjust for the probability of selection. Analysis of the weighted data was conducted using SAS Enterprise. Differences in prevalence by sex, age and other sociodemographic characteristics were described using Wald Chi Squared Tests, accounting for the complex survey design. Subsequently, the age-adjusted prevalence estimates derived in this study were extrapolated to the Nigerian childhood population estimate to estimate the prevalence of childhood surgical conditions across the country.

## Results

SOSAS-NST was administered to 856 households. One hundred fifteen children were excluded due to age greater than 18 years or missing information. Among the 1883 children included in the study, the median age of respondents was 8.0 (IQR: 8.0, range: 5 months– 18 years). Eighty-one surgical conditions were identified, the most common of which being umbilical hernias (20), inguinal hernias (13), and wound injuries to the extremities (9). The prevalence of common congenital conditions is shown in **Table 1**. The prevalence of umbilical hernias was 85 per 10,000 children in Ikorodu (95% CI: 47, 123), and the prevalence of inguinal hernias was 61 per 10,000 (95% CI: 34, 88). The prevalence of hydroceles and undescended testes were comparable at 22 and 26 per 10,000 children, respectively. Hydrocephalus and cleft palate were also observed in the sample. There were also a range of conditions reported that were unspecified, including wound injuries, burns, masses and other congenital and acquired conditions.

Socio-demographic characteristics of children and parents by surgical conditions in the study are reported in **Table 2**. Younger parent age, younger child age and smaller family size were all associated with increased prevalence of surgical conditions. Village type, parent

**Table 1. Prevalence of pediatric surgical conditions per 10,000 children.**

| | | Number of Cases in Cohort (N = 1883) | Prevalence per 10,000 | 95% Confidence Interval | |
|---|---|---|---|---|---|
| **Identified Conditions** | | | | | |
| Hydrocephalus | | 1 | 2.3 | 0 | 6.6 |
| Cleft Palate | | 1 | 4.3 | 0 | 12.5 |
| Undescended Testes | | 5 | 26.1 | 8.4 | 43.8 |
| Umbilical Hernia | | 20 | 85.1 | 47.3 | 122.9 |
| Inguinal Hernia | | 13 | 61.1 | 33.8 | 88.4 |
| Hydrocele | | 5 | 21.7 | 4.0 | 39.4 |
| Any Surgical Condition | | 40 | 183.3 | 134.3 | 232.3 |
| **Other Conditions** | | | | | |
| Abdomen | Abdominal distention | 2 | 12.2 | 0 | 31.2 |
| | Burn | 2 | 4.6 | 0 | 13.2 |
| Groin | Congenital deformity | 1 | 6.2 | 0 | 16.5 |
| Facial | Acquired deformity | 3 | 20.6 | 1.0 | 40.1 |
| | Congenital deformity | 1 | 4.3 | 0 | 12.5 |
| | Injury | 3 | 6.9 | 0 | 19.8 |
| | Mass | 3 | 4.7 | 0 | 12.8 |
| | Wound Injury | 1 | 4.2 | 0 | 12.5 |
| Extremities | Acquired deformity | 3 | 16.6 | 0 | 37.4 |
| | Congenital deformity | 3 | 13.8 | 1.0 | 26.5 |
| | Injury | 3 | 9.0 | 0 | 19.3 |
| | Mass | 1 | 6.2 | 0 | 16.5 |
| | Wound Injury | 9 | 57.3 | 0.1 | 114.6 |
| | Burn | 1 | 6.2 | 0 | 16.5 |

education, literacy, tribe, parent employment and child sex did not seem to be correlated to the prevalence of surgical conditions.

The age-adjusted prevalence of all reported surgical conditions in Ikorodu was 352 per 10,000 children. The age-specific prevalence of surgical conditions in children less than 1 year is 480 in 10,000, compared to 344 in children 1–2 years, 295 in children 3–5 years, 406 in children 6–11 years, 218 in children 12–14 years, and 379 in children 15–18 years.

## Discussion

This study is the first from Nigeria to estimate the prevalence of surgically correctable diseases among children based entirely on community data. Following the landmark work of the LCoGS, it has been advocated that for the huge gaps in surgical capacity to be bridged, countries will need to have workable National Surgical Obstetric and Anaesthetic Plans (NSOAPs) if the Sustainable Development Goals (SDGs) and LCoGS 2030 goals are to be achieved [1]. For a robust NSOAP, accurate data is essential for informing the correct policy formulations. However, there is very limited population-based data from LMICs including Nigeria, and most of the policies have been based on extrapolations from hospital-based studies which are fraught with many inaccuracies. Furthermore, most National Surgical Health Plans have so far not addressed children's surgery [15]. This study provides much-needed data on the burden of paediatric surgical diseases for policy makers in our country.

In our study, the overall prevalence of both congenital and acquired surgically correctable conditions among children in the mixed urban-rural population of Ikorodu was 352 per 10,000 or 3.52%. The Nigeria Population Commission estimates the current population of the

**Table 2. Characteristics associated with childhood surgical conditions (Any reported condition).**

| Characteristics | | Total (N = 1883) | | No Condition (N = 1808) | | Condition (N = 75) | | P value |
|---|---|---|---|---|---|---|---|---|
| Parent Age, Median (SD) | | 38.0 | 15.0 | 39.0 | 15.0 | 36.0 | 18.0 | <0.0001 |
| Village type | Rural | 1500 | 80% | 1446 | 80% | 54 | 72% | 0.96 |
| | Urban | 383 | 20% | 362 | 20% | 21 | 28% | |
| Parent Education | No formal education | 246 | 13% | 239 | 13% | 7 | 9% | 0.36 |
| | Primary school | 377 | 20% | 366 | 20% | 11 | 15% | |
| | Junior secondary | 122 | 6% | 119 | 7% | 3 | 4% | |
| | Senior secondary | 863 | 46% | 817 | 45% | 46 | 61% | |
| | Tertiary school | 249 | 13% | 244 | 13% | 5 | 7% | |
| | Graduate degree | 17 | 1% | 15 | 1% | 2 | 3% | |
| Parent Literacy | No | 249 | 13% | 416 | 23% | 13 | 17% | 0.81 |
| | Yes | 1434 | 76% | 1374 | 76% | 60 | 80% | |
| Family Size, Median (SD) | | 6.0 | 3.0 | 6.0 | 3.0 | 5.0 | 3.0 | <0.0001 |
| Tribe | Yoruba | 1652 | 88% | 1583 | 88% | 69 | 92% | 0.20 |
| | Igbo | 130 | 7% | 127 | 7% | 3 | 4% | |
| | Hausa | 19 | 1% | 17 | 1% | 1 | 1% | |
| | Other | 73 | 4% | 73 | 4% | 1 | 1% | |
| Parent Employment | Unemployed | 151 | 8% | 145 | 8% | 6 | 8% | 0.38 |
| | Home maker | 37 | 2% | 35 | 2% | 2 | 3% | |
| | Domestic helper | 42 | 2% | 42 | 2% | 0 | 0% | |
| | Farmer | 22 | 1% | 22 | 1% | 0 | 0% | |
| | Self-employed | 1403 | 75% | 1349 | 75% | 54 | 72% | |
| | Government employee | 105 | 6% | 101 | 6% | 4 | 5% | |
| | Non-government employee | 108 | 6% | 100 | 6% | 8 | 11% | |
| | Student | 7 | 0% | 7 | 0% | 0 | 0% | |
| Child Sex | Female | 903 | 48% | 870 | 48% | 33 | 44% | 0.50 |
| | Male | 977 | 52% | 935 | 52% | 42 | 56% | |
| Child Age, Median (SD) | | 8.0 | 8.0 | 8.0 | 8.0 | 7.0 | 7.0 | 0.0002 |

Percentages may not add up to 100% due to missing data.

country to be about 198 million [16]. In keeping with most other LMICs, children comprise a large proportion of Nigeria's population. [17] The 3.52% prevalence rate crudely translates into 2.9 million children across the country with a surgically correctable condition, approximately one third of which were due to congenital conditions.

The commonest congenital conditions encountered in this study are umbilical hernias, inguinal hernias, hydroceles and undescended testes, with prevalence estimates per 10,000 children of 85, 61, 26 and 22 respectively. Umbilical hernias are common in our setting and has been estimated to be about 80 times higher than the prevalence in HICs [18–19]. The prevalence of umbilical hernia has been estimated to be between 10% and 85%. [20] While most of these resolves spontaneously, some persist and require surgical correction. Health education during antenatal care on the proper care of the cord needs to be emphasised.

Other community-based studies done in Europe and China have reported prevalence rates of 0.4–1.4% for inguinal hernias, and our study prevalence rate of 0.6% fits well within this range. [21–22]. This finding is supported by other workers that have found inguinal hernias as the commonest prevalent conditions in the community. [23] Previous SOSAS surveys have found a higher incidence of "inguinal swellings". [6] However, in those studies, the prevalence

was in the whole population including adults and children, and no physical examination was conducted to confirm the reported conditions—hence potentially leading to over-reporting of other groin swellings such as lymphadenopathy, hydroceles and femoral hernias as inguinal hernias. This study, on the other hand, included a physical examination by a surgeon to confirm the diagnosis, with expected increase in diagnostic accuracy. This is the first implementation of SOSAS to introduce physical examination to help confirm subject reports of surgical conditions, with the hope to generate more accurate, even if lower, prevalence estimates compared to previous studies.

The commonest acquired surgically correctable conditions in this survey are injuries and soft tissue abscesses. Injuries remain a very significant public health problem among children and youth worldwide. They are responsible for more deaths than HIV/AIDS, tuberculosis and malaria put together and affect mostly children and young people. [24] Apart from mortality, the residual morbidity of injuries, expressed in disability-adjusted life years (DALYs), is very significant. It has been estimated that 90% of the global injury burden occurs in LMICs. [25] Our study suggests that over 600,000 Nigerian children may be having one or more disabilities or deformities from previous injuries–and this is likely an underestimate considering the impact of violence and insecurity in different parts of the country. These sobering statistics call for concerted efforts to apply known, cost-effective interventions in the prevention and management of injuries in children. [24]

In a similar vein, our study has reported a high prevalence of soft tissue infections and abscesses. These conditions may be caused by minor injuries or direct infectious agents, but are likely also related to poor hygiene and reluctance to seek prompt medical attention. A well-structured health education program on sound personal hygiene as well as school-based health education programs may be necessary to address this public health issue.

This study also identifies a higher prevalence of paediatric surgical diseases in urban and more educated families. These population trends could be artefactual, due to conditions being reported less on in rural populations or those with less education.

Overall, this study has revealed a high burden of paediatric surgical diseases, that easily overwhelms the available workforce capacity. As an immediate measure, there may be need for "task-shifting" to allow general surgeons and urologists with competence in the surgical care of children to treat the simple straightforward and highly prevalent conditions such as hernias, undescended testes and hydroceles. Another immediate interventional measure will be the setting up of "surgical camps" which can be organized through the local government areas or local community development areas where children with surgical conditions are screened and operated as day cases over a 1–2 week period with surgeons and technical support staff camped in the community for this period to fill in the gap of those with unmet surgical needs. As intermediate measures to increase paediatric surgical competence among surgical trainees, the training colleges should be encouraged to adopt rotations in paediatric surgery as a compulsory (as opposed to the current elective model) requirement for all surgeons for this purpose and increase the rotation to 6 months instead of the current 3 months. This will allow most surgeons irrespective of specialty, to take care of simple paediatric surgical cases. In the long term, a NSOAP that incorporates paediatric surgery, allocates resources to training of more paediatric surgeons including attractive incentives and continued advocacy and engagement of government and all stakeholders will be the appropriate plan to address the shortages.

Based on a recent estimate of 100 practising paediatric surgeons in Nigeria [26], the implication is that only one paediatric surgeon is available to treat approximately 29,000 paediatric patients who currently have surgically correctable conditions.

As far back as 20 years ago, the United States have had a ratio of 10 paediatric surgeons to 1 million children [27]. Even if Nigeria will make half of that an aspirational goal, with about 82

million children, the country will require a minimum of 410 paediatric surgeons which suggests a deficit of 310 paediatric surgeons. While this may be a tall order, it is important that this grim reality be made known to policy makers and other stakeholders when formulating policies that will affect children's surgery, in order to recognize immediate, intermediate and long-term interventions that may be helpful in addressing the huge gaps.

Our study has several limitations, some of which have been highlighted earlier. In the first place, we studied a small community in a relatively well-resourced region of the country in terms of health personnel. Hence, a more accurate picture would be presented if the study were done nationwide. However, the cost of performing the study nationally, in terms of human and material resources, would be prohibitive. Another limitation of the study is that some of the conditions seen could not be diagnosed without further investigations and that was outside the scope of the present study. Hence patients with various intra-abdominal, joint and limb, and head and neck masses and swellings could not be further categorized in this study. This is a general limitation of the SOSAS instrument which is basically focused on adults and vague characterisation of pathologies in different parts of the body.

In conclusion, our study defined the prevalence of common congenital and acquired surgically correctable conditions in a mixed rural-urban area of Nigeria, then extrapolated the results nationally, highlighting a huge gap in paediatric surgical workforce in the country. It is hoped that the suggested recommendations can change the narrative and ultimately improve the surgical care provision of children in resource-constrained settings.

## Supporting information

**S1 File. Surgeons' Overseas Assessment of Surgical Needs–Nigeria Survey Tool (SOSAS-NST) questionnaire.**
(PDF)

## Acknowledgments

The authors are grateful to the following Field Enumerators for their assistance: Dr. Victor B. Nwinee, Dr. Adenuga Oyeniyi, Dr. Akinsanya Oluwatunmise, Dr. Akinpelu Akintunde, Dr. Chilaka Grace A., Dr. Murphy-Akpieyi Ofeoritse, Dr. Margaret Anurika Reuben, Dr. Babalola Hannah F., Dr. Ezinwa E.U., Ugwuadu, Dr. Olorunda Oluwaseyi C., Dr. Peculiar Ibeh, Dr. John Daikpor, Dr. Oluwakemi Adetayo, Dr. Kenneth Uche Onyekachi, Dr. Lawal Olusola, Dr. Yetunde Atiba, Dr. Seun S. Odusanmi, Dr. Ajayi Oladele Samuel, Dr. Obanubi Nicholas O, Dr. Kayode D. Oluremi.

## Author Contributions

**Conceptualization:** Adesoji O. Ademuyiwa, Tinuola O. Odugbemi, Christopher O. Bode, Olumide A. Elebute, Felix M. Alakaloko, Olufemi Bankole, Tu Tran, Ayomide Makanjuola, Shailvi Gupta, Ewen M. Harrison, Dan Poenaru, Benedict C. Nwomeh.

**Data curation:** Adesoji O. Ademuyiwa, Tinuola O. Odugbemi, Olumide A. Elebute, Felix M. Alakaloko, Eyitayo O. Alabi, Olufemi Bankole, Oluwaseun Ladipo-Ajayi, Justina O. Seyi-Olajide, Babasola Okusanya, Ogechi Abazie, Iyabo Y. Ademuyiwa, Ayomide Makanjuola, Riinu Ots, Ewen M. Harrison.

**Formal analysis:** Amanda Onwuka, Tu Tran, Ayomide Makanjuola.

**Funding acquisition:** Adesoji O. Ademuyiwa, Tinuola O. Odugbemi.

**Methodology:** Adesoji O. Ademuyiwa, Tinuola O. Odugbemi, Christopher O. Bode, Olumide A. Elebute, Felix M. Alakaloko, Eyitayo O. Alabi, Olufemi Bankole, Oluwaseun Ladipo-Ajayi, Justina O. Seyi-Olajide, Babasola Okusanya, Ogechi Abazie, Iyabo Y. Ademuyiwa, Amanda Onwuka, Tu Tran, Ayomide Makanjuola, Shailvi Gupta, Riinu Ots, Ewen M. Harrison, Dan Poenaru, Benedict C. Nwomeh.

**Project administration:** Adesoji O. Ademuyiwa, Tinuola O. Odugbemi, Olumide A. Elebute, Felix M. Alakaloko, Olufemi Bankole, Ayomide Makanjuola, Dan Poenaru, Benedict C. Nwomeh.

**Resources:** Adesoji O. Ademuyiwa.

**Supervision:** Adesoji O. Ademuyiwa, Tinuola O. Odugbemi, Christopher O. Bode, Olumide A. Elebute, Felix M. Alakaloko, Eyitayo O. Alabi, Olufemi Bankole, Oluwaseun Ladipo-Ajayi, Justina O. Seyi-Olajide, Babasola Okusanya, Ogechi Abazie, Iyabo Y. Ademuyiwa, Amanda Onwuka, Ayomide Makanjuola, Ewen M. Harrison, Dan Poenaru, Benedict C. Nwomeh.

**Validation:** Adesoji O. Ademuyiwa, Amanda Onwuka, Tu Tran, Ayomide Makanjuola, Benedict C. Nwomeh.

**Writing – original draft:** Adesoji O. Ademuyiwa, Tinuola O. Odugbemi, Dan Poenaru, Benedict C. Nwomeh.

**Writing – review & editing:** Adesoji O. Ademuyiwa, Tinuola O. Odugbemi, Christopher O. Bode, Olumide A. Elebute, Felix M. Alakaloko, Eyitayo O. Alabi, Olufemi Bankole, Oluwaseun Ladipo-Ajayi, Justina O. Seyi-Olajide, Babasola Okusanya, Ogechi Abazie, Iyabo Y. Ademuyiwa, Amanda Onwuka, Tu Tran, Ayomide Makanjuola, Shailvi Gupta, Riinu Ots, Ewen M. Harrison, Benedict C. Nwomeh.

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
