## [Decision Letter · Decision Letter 0]

2 Aug 2019

PONE-D-19-18561

Prevalence of surgically correctable conditions among children in a mixed urban-rural community in Nigeria using the SOSAS survey tool: implications for paediatric surgical capacity-building.

PLOS ONE

Dear Dr Ademuyiwa,

Thank you for submitting your manuscript to PLOS ONE. After careful consideration, we feel that it has merit but does not fully meet PLOS ONE’s publication criteria as it currently stands. Therefore, we invite you to submit a revised version of the manuscript that addresses the points raised during the review process.

We would appreciate receiving your revised manuscript by Sep 16 2019 11:59PM. To enhance the reproducibility of your results, we recommend that if applicable you deposit your laboratory protocols in protocols.io, where a protocol can be assigned its own identifier (DOI) such that it can be cited independently in the future. For instructions see: http://journals.plos.org/plosone/s/submission-guidelines#loc-laboratory-protocols

We look forward to receiving your revised manuscript.

Kind regards,

Akshay Chauhan

Academic Editor

PLOS ONE

Journal Requirements:

a) Did participants provide their written or verbal informed consent to participate in this study?

Reviewers' comments:

Reviewer's Responses to Questions

**Comments to the Author**

1. Is the manuscript technically sound, and do the data support the conclusions?

Reviewer #1: Yes

Reviewer #2: Partly

2. Has the statistical analysis been performed appropriately and rigorously? 

Reviewer #1: I Don't Know

Reviewer #2: Yes

3. Have the authors made all data underlying the findings in their manuscript fully available?

Reviewer #1: No

Reviewer #2: Yes

4. Is the manuscript presented in an intelligible fashion and written in standard English?

Reviewer #1: Yes

Reviewer #2: Yes

5. Review Comments to the Author

Reviewer #1: The study aims to determine the prevalence of common surgical conditions in children living in a specific community in Nigeria. For this, a household survey was completed using a globally accepted questionnaire. The authors highlight the limitations in the current study's data in Nigeria due to the fact that it is mainly provided by hospital-based studies. The authors claim that community-based prevalence studies may provide a better estimation of surgical need. The ultimate goal is to improve the provision of surgical care for children in the community through implementing proper resource allocation and prioritization.

According to The Lancet Commission on Global Surgery, 5 billion people (over half of the world's population) lack access to affordable surgical care1,2. Each year, 313 million surgical procedures are performed worldwide, but only 6% of them are done in the low- and middle-income countries (LMICs)1. 2/3 of the children around the world lack access to surgical care and comprise up to half of the population in the LMICs. There is a considerable mismatch between the pediatric population and the number of pediatric surgeons3.

Although little attention has been directed towards addressing this problem historically, international organizations have been working together in order to promote improvements and changes to this panorama3. Guidelines and questionnaires have been created for data collection and analysis. It also has been suggested that hospital-level collection data in LMICs may be deficient4. Hence, a comprehensive national assessment should be extended to the community level5.

Therefore, the idea of determining the prevalence of common surgical conditions in children through a community-based study in a specific area in Nigeria is valuable and relevant.

The current study contains a well-structured methodology. The Surgeons OverSeas Assessment of Surgical Need (SOSAS) that was designed using the Demographic and Health Surveys (DHS) was used, which is the most comprehensive source for population-based health statistics in developing countries5. The data collection technique is notable, and the inclusion of a visual portfolio and physical exam at the end of each interview may generate more accurate data compared to other studies.

The introduction and discussion have a good flow and readability. The study used a comprehensive and solid scientific literature list, including similar studies from Sub-Saharan Africa countries5,6.

However, some minor issues should be addressed:

MINOR ISSUES

1. Line 392: “Hence, our data is likely to present a better picture than if the study were done nationwide”. If the study was done nationwide there would be no extrapolation, and the data would be more accurate7. However, as was pointed out (line 394), it would be economically prohibitive.

a. Recommendation: changing the sentence for “better picture if the study were done nationwide” would solve the problem.

2. Line 272: “Assuming a similar prevalence value across the entire country…”.

a. Recommendation: this assumption would fit better in the discussion linked to the extrapolation data idea at line 308-309 (“The 3.52% prevalence rate crudely translates into 2.9 million children...”).

3. Line 176: SOSAS modification is cited

a. Recommendation: including the modification performed on the questionnaire SOSAS in “data collection tool” would give the study more transparency.

REFERENCE:

1. Meara JG, Leather AJM, Hagander L, et al. Global Surgery 2030: Evidence and solutions for achieving health, welfare, and economic development. Lancet. 2015;386(9993):569-624. doi:10.1016/S0140-6736(15)60160-X

2. Kim, JY. Opening address to the inaugural ‘‘The Lancet Commission on Global Surgery” meeting. The World Bank. Jan 17, 2014. Boston, MA, USA. http://www.globalsurgery.info/wp-content/ uploads/2014/01/Jim-Kim-Global-Surgery-Trraanscribed.pdf

3. Goodman LF, St-Louis E. The Global Initiative for Children’s Surgery: Optimal Resources for Improving Care. Artic Eur J Pediatr Surg. 2017. doi:10.1055/s-0037-1604399

4. O’Brien MJ, Whitaker RC. The role of community-based participatory research to inform local health policy: A case study. J Gen Intern Med. 2011. doi:10.1007/s11606-011-1878-3

5. Petroze RT, Groen RS, Niyonkuru F, et al. Estimating operative disease prevalence in a low-income country: Results of a nationwide population survey in Rwanda. Surg (United States). 2013;153(4):457-464. doi:10.1016/j.surg.2012.10.001

6. Fuller AT, Butler EK, Tran TM, et al. Surgeons OverSeas Assessment of Surgical Need (SOSAS) Uganda: Update for Household Survey. World J Surg. 2015. doi:10.1007/s00268-015-3191-5

7. Mock CN, Donkor P, Gawande A, et al. Essential surgery: Key messages from Disease Control Priorities, 3rd edition. Lancet. 2015. doi:10.1016/S0140-6736(15)60091-5

Reviewer #2: 1) Was your study rural only? In your methods section, you describe the incorporation of both rural and urban wards in your sampling strategy; however this should be underscored in your description here.

2)Line 134: Define ‘adequate community engagement

3)Line 148: I would be cautious in including subjective definitions of household here, unless this is supported by the literature (if so, please cite). A group of people who slept under the same room the night prior to the interview would be a sufficient definition.

4)Line 164: What is the average household size and where did you obtain this data (cite).

5)Line 170: I suspect this needs to be corrected to ‘One street/settlement’

6)Line 172: How did you identify residential buildings? Were informal settlements included in this process?

7) Lines 176 and following: Cite the studies in which SOSAS was first introduced. How was the SOSAS modified for this study?

8) Line 241: please be more specific about the type of regression analyses used

9) Line 272: Was the extrapolated prevalence adjusted for age demographics of children and parents, as well as family size, since those variables were correlated with the prevalence of the studied surgical conditions?

10) Was there any assessment/inclusion of surgical diagnoses that had already been addressed in the population studied? If so, did this differ between Urban/Rural or by other variables?

11) Lines 314-317: The discussion of UH prevalence seems unnecessary given that your study prevalence is only 85 in 10,000 (0.85%).

12) Lines 338-350 This paragraph would benefit from review by an editor-the sentence structure and language is cumbersome and confusing

13) Lines 342-343: please cite pediatric trauma/injury burden data appropriately

14) Line 351 and following: Have you considered other sources of abscesses and soft tissue infections, such as minor trauma, infectious disease, etc?

15) Be consistent in use of pediatric surgery vs childhood surgery

16) Line 379-383: US data suggests the need for 1 pediatric surgeon to 100,000 pediatric patients. You are suggesting Nigeria has a higher proportion of pediatric surgeon to patients (1:29,000), is this correct or is the 100:82 million correct?

6. PLOS authors have the option to publish the peer review history of their article (what does this mean?). If published, this will include your full peer review and any attached files.

Reviewer #1: No

Reviewer #2: No

---

## [Author Response · Author response to Decision Letter 0]

19 Aug 2019

Reviewer #1: The study aims to determine the prevalence of common surgical conditions in children living in a specific community in Nigeria. For this, a household survey was completed using a globally accepted questionnaire. The authors highlight the limitations in the current study's data in Nigeria due to the fact that it is mainly provided by hospital-based studies. The authors claim that community-based prevalence studies may provide a better estimation of surgical need. The ultimate goal is to improve the provision of surgical care for children in the community through implementing proper resource allocation and prioritization.

According to The Lancet Commission on Global Surgery, 5 billion people (over half of the world's population) lack access to affordable surgical care1,2. Each year, 313 million surgical procedures are performed worldwide, but only 6% of them are done in the low- and middle-income countries (LMICs)1. 2/3 of the children around the world lack access to surgical care and comprise up to half of the population in the LMICs. There is a considerable mismatch between the pediatric population and the number of pediatric surgeons3.

Although little attention has been directed towards addressing this problem historically, international organizations have been working together in order to promote improvements and changes to this panorama3. Guidelines and questionnaires have been created for data collection and analysis. It also has been suggested that hospital-level collection data in LMICs may be deficient4. Hence, a comprehensive national assessment should be extended to the community level5.

Therefore, the idea of determining the prevalence of common surgical conditions in children through a community-based study in a specific area in Nigeria is valuable and relevant.

The current study contains a well-structured methodology. The Surgeons OverSeas Assessment of Surgical Need (SOSAS) that was designed using the Demographic and Health Surveys (DHS) was used, which is the most comprehensive source for population-based health statistics in developing countries5. The data collection technique is notable, and the inclusion of a visual portfolio and physical exam at the end of each interview may generate more accurate data compared to other studies.

The introduction and discussion have a good flow and readability. The study used a comprehensive and solid scientific literature list, including similar studies from Sub-Saharan Africa countries5,6.

AUTHOR RESPONSE: The authors are grateful to the reviewers for their kind comments and thorough review. The minor issues raised by this reviewer will be addressed below

However, some minor issues should be addressed:

MINOR ISSUES

1. Line 392: “Hence, our data is likely to present a better picture than if the study were done nationwide”. If the study was done nationwide there would be no extrapolation, and the data would be more accurate7. However, as was pointed out (line 394), it would be economically prohibitive.

a. Recommendation: changing the sentence for “better picture if the study were done nationwide” would solve the problem.

AUTHOR'S RESPONSE: Recommendation taken and correction effected (lines 409 – 410)

2. Line 272: “Assuming a similar prevalence value across the entire country…”.

a. Recommendation: this assumption would fit better in the discussion linked to the extrapolation data idea at line 308-309 (“The 3.52% prevalence rate crudely translates into 2.9 million children...”).

AUTHOR'S RESPONSE:Thank you for this recommendation. This has been addressed comprehensively – please see response to reviewer 2, item 9 below

3. Line 176: SOSAS modification is cited

a. Recommendation: including the modification performed on the questionnaire SOSAS in “data collection tool” would give the study more transparency.

AUTHORS' RESPONSE: The modified SOSAS instrument – the SOSAS – NST (Nigeria Survey Tool) will be deposited as a supplementary tool for reference. This information is in line 551- 553 after Reference section of the manuscript.

Reviewer #2: 1) Was your study rural only? In your methods section, you describe the incorporation of both rural and urban wards in your sampling strategy; however this should be underscored in your description here.

AUTHORS' RESPONSE: Ikorodu is a mixed urban and rural agrarian community. This is reflected in lines 130 -131

2)Line 134: Define ‘adequate community engagement

AUTHORS' RESPONSE: By adequate community participation, we mean, from the community leaders, women and youth organizations, market leaders, ordinary community men and women were adequately briefed about the study before its commencement and when randomized into the study, actively participated through the volunteering to answer the questionnaires and get physically examined. We had less than 5% of persons approached who declined to be part of the study – lines 135 - 142

3)Line 148: I would be cautious in including subjective definitions of household here, unless this is supported by the literature (if so, please cite). A group of people who slept under the same room the night prior to the interview would be a sufficient definition.

AUTHORS' RESPONSE: The appropriate reference has been cited for this definition. Line 159

4)Line 164: What is the average household size and where did you obtain this data (cite).

AUTHORS' RESPONSE: Average house hold size is 4.6 or approximately 5 (National Population Commission and ICF International. Nigeria Demographic and Health Survey 2013. Abuja, Nigeria, and Rockville, Maryland, USA: NPC and ICF International. 2014:pp19 (line 172)

https://dhsprogram.com/pubs/pdf/FR293/FR293.pdf )

5)Line 170: I suspect this needs to be corrected to ‘One street/settlement’

AUTHORS' RESPONSE: Suggested correction has been made – line 178

6)Line 172: How did you identify residential buildings? Were informal settlements included in this process?

AUTHORS' RESPONSE: Residential buildings were identified by local community Field Officers who accompanied our team and informal settlements were included 

7) Lines 176 and following: Cite the studies in which SOSAS was first introduced. How was the SOSAS modified for this study?

AUTHORS' RESPONSE: The SOSAS study was first carried out in Sierra Leone. The references have been cited in line 19778. The full questionnaire of the modified SOSAS – SOSAS- NST (Nigeria Survey Tool) is now deposited as supplementary file for reference purposes. The summary of the modifications are in the paragraph from lines 201 – 213

8) Line 241: please be more specific about the type of regression analyses used

AUTHORS' RESPONSE: Thank you for identifying this. The primary analysis method for this study were Wald Chi Squared tests. Previous versions of the manuscript included regression analyses but they were not included in the final version. This sentence in the methods has been revised to reflect only the analyses presented. “Differences in prevalence by sex, age and other sociodemographic characteristics were described using Wald Chi Squared Tests, accounting for the complex survey design” (line 249)

9) Line 272: Was the extrapolated prevalence adjusted for age demographics of children and parents, as well as family size, since those variables were correlated with the prevalence of the studied surgical conditions?

AUTHORS' RESPONSE: The prevalence estimate was age-adjusted, but not adjusted for parent age or family size as population level data for those characteristics is not attainable. We have included age-specific prevalence rates as well. (lines 282 – 285)

10) Was there any assessment/inclusion of surgical diagnoses that had already been addressed in the population studied? If so, did this differ between Urban/Rural or by other variables?

AUTHORS' RESPONSE: In this study, we requested families report any condition (addressed or unaddressed) that occurred in the prior 12 months for each child. Eleven of 81 (13%) of conditions had been treated surgically at the time of the survey. We intend to describe patterns in healthcare utilization and barriers to surgical care in a subsequent paper and have not described those results here

11) Lines 314-317: The discussion of UH prevalence seems unnecessary given that your study prevalence is only 85 in 10,000 (0.85%).

AUTHORS' RESPONSE: The authors are of the opinion that being the commonest congenital condition, it is important to discuss its prevalence nonetheless as most quoted figures are hospital based rather than community derived figures.

12) Lines 338-350 This paragraph would benefit from review by an editor-the sentence structure and language is cumbersome and confusing

AUTHORS' RESPONSE: The paragraph has been edited (lines 342 – 346)

13) Lines 342-343: please cite pediatric trauma/injury burden data appropriately

AUTHORS' RESPONSE: The sentence has been appropriately cited (lines 356 -360)

14) Line 351 and following: Have you considered other sources of abscesses and soft tissue infections, such as minor trauma, infectious disease, etc?

AURHORS' RESPONSE: Consideration has been given to other causes ( lines 368 -369)

15) Be consistent in use of pediatric surgery vs childhood surgery

AUTHORS' COMMENTS: Thanks for the comment, correction effected in line 392

16) Line 379-383: US data suggests the need for 1 pediatric surgeon to 100,000 pediatric patients. You are suggesting Nigeria has a higher proportion of pediatric surgeon to patients (1:29,000), is this correct or is the 100:82 million correct?

AUTHORS' COMMENTS: The US has 1 paediatric surgeon to 100,000 children (total population) of which a percentage say <5% have surgical disease i.e. approx. <5000 children with surgical conditions compared to Nigeria where it is 1 paediatric surgeon to 29000 children with surgical conditions

---

## [Editor Report · Decision Letter 1]

23 Sep 2019

Prevalence of surgically correctable conditions among children in a mixed urban-rural community in Nigeria using the SOSAS survey tool: implications for paediatric surgical capacity-building.

PONE-D-19-18561R1

Dear Dr. Ademuyiwa,

We are pleased to inform you that your manuscript has been judged scientifically suitable for publication and will be formally accepted for publication once it complies with all outstanding technical requirements.

With kind regards,

Akshay Chauhan

Academic Editor

PLOS ONE

Additional Editor Comments (optional):

Dear Dr Adesoji O Ademuyiwa, I am pleased to inform you that your manuscript titled "Prevalence of surgically correctable conditions among children in a mixed urban-rural community in Nigeria using the SOSAS survey tool: implications for paediatric surgical capacity-building" has been accepted for publication in PLOS ONE journal. The decision was based on double blinded review process. Congratulations.
---

## [Editor Report · Acceptance letter]

4 Oct 2019

PONE-D-19-18561R1 

Prevalence of surgically correctable conditions among children in a mixed urban-rural community in Nigeria using the SOSAS survey tool: implications for paediatric surgical capacity-building. 

Dear Dr. Ademuyiwa:

I am pleased to inform you that your manuscript has been deemed suitable for publication in PLOS ONE. Congratulations! Your manuscript is now with our production department. 

With kind regards,

on behalf of

Dr. Akshay Chauhan 

Academic Editor

PLOS ONE